# Functional analysis of CASK transcript variants expressed in human brain

**Debora Tibbe**[1], **Yingzhou Edward Pan**[1], **Carsten Reißner**[2], **Frederike L. Harms**[1], **Hans-Jürgen Kreienkamp**[1] *

**1** Institut für Humangenetik, Universitätsklinikum Hamburg-Eppendorf, Hamburg, Germany, **2** Institut für Anatomie und Molekulare Neurobiologie, Westfälische Wilhelms-Universität Münster, Münster, Germany

* kreienkamp@uke.de

## Abstract

The calcium-/calmodulin dependent serine protein kinase (CASK) belongs to the membrane-associated guanylate kinases (MAGUK) family of proteins. It fulfils several different cellular functions, ranging from acting as a scaffold protein to transcription control, as well as regulation of receptor sorting. CASK functions depend on the interaction with a variety of partners, for example neurexin, liprin-α, Tbr1 and SAP97. So far, it is uncertain how these seemingly unrelated interactions and resulting functions of CASK are regulated. Here, we show that alternative splicing of CASK can guide the binding affinity of CASK isoforms to distinct interaction partners. We report seven different variants of CASK expressed in the fetal human brain. Four out of these variants are not present in the NCBI GenBank database as known human variants. Functional analyses showed that alternative splicing affected the affinities of CASK variants for several of the tested interaction partners. Thus, we observed a clear correlation of the presence of one splice insert with poor binding of CASK to SAP97, supported by molecular modelling. The alternative splicing and distinct properties of CASK variants in terms of protein-protein interaction should be taken into consideration for future studies.

## Introduction

Alternative splicing of pre-mRNA is an essential concept to increase the diversity of gene expression and for expanding the coding power of genomes [1]. Through inclusion and exclusion of exons, as well as the selection of alternative 5´- and 3´- splice sites, multiple protein isoforms can be generated from a single gene [2–4]. In the human genome, alternative splicing enables the production of more than 90,000 different proteins from about 20,000 protein-coding genes. Alternative splicing occurs in more than 90% of human pre-mRNAs [5, 6]. Cell-type specific alternative splicing plays an important role for cellular differentiation and development, especially in the CNS [7]. Cell type-specific alternative splicing programs are extensively regulated across different brain regions and developmental stages, contributing to the complexity of specialized cell-populations in the human brain [8, 9].

Membrane-associated guanylate kinases (MAGUKs) form a family of scaffold proteins that occur enriched at various intercellular junctions, including synapses [10]. MAGUKs form

**Data Availability Statement:** All relevant data are within the paper and its Supporting Information files.

**Funding:** This work was supported by a grant from the Deutsche Forschungsgemeinschaft (Kr 1321/7-

1, to H.-J.K.). F.L.H. was supported by the Research Promotion Fund of the Faculty of Medicine (FFM) of the University Medical Center Hamburg-Eppendorf and by a grant from the Deutsche Forschungsgemeinschaft (KU 1240/10-1). The funders had no role in study design, data collection and analysis, decision to publish, or preparation of the manuscript.

**Competing interests:** The authors have declared that no competing interests exist.

scaffolds for the active zone at the presynaptic site and the postsynaptic density by bringing together membrane receptors, ion channels, signaling proteins and cytoskeletal proteins to form larger protein complexes [11]. In doing so, MAGUKs perform important functions in cell adhesion and intercellular signal transduction, as well as in synaptic development and plasticity [11–13]. The PSD-95 family members of MAGUK proteins, PSD-95, PSD-93, SAP97 and SAP102 are among the best characterized MAGUKs. The pre-mRNAs of all members of this family can undergo alternative splice events, which have important consequences for transport and synaptic function of the MAGUK proteins [12, 14, 15].

The calcium/calmodulin-dependent serine protein kinase (CASK) also belongs to the MAGUK protein family [16]. The human *CASK* gene is located on the X chromosome; patients with mutations in *CASK* exhibit X-linked intellectual disability (XLID) [17] and microcephaly with pontine and cerebellar hypoplasia (MICPCH) [18, 19]. In neurons, the CASK protein is localized in the nucleus, the soma, and close to pre- and postsynaptic plasma membranes. CASK fulfils its various functions by associating with a multitude of interaction partners [14]. CASK has the typical MAGUK interaction domains, including the PDZ domain [post synaptic density protein (PSD-95), Drosophila disc large tumor suppressor (Dlg1), Zonula occludens-1 protein (ZO-1)], SH3 (Src homology 3), GK (guanylate kinase) and two L27 (MAGUK LIN-2 + LIN-7) domains [10, 13, 20]. The N-terminus of CASK is formed by a calcium/calmodulin dependent kinase (CaMK) domain that is not present in any other MAGUK [20].

The PDZ domain of CASK interacts with the intracellular C-terminus of the presynaptic cell adhesion molecule neurexin (Nrxn) [16]. Neurexin isoforms bind neuroligins, cell adhesion molecules of the postsynapse [21], thus forming a trans-synaptic complex which contributes to synapse formation and synaptic plasticity [20]. CASK interacts with several additional presynaptic proteins; thus it is involved in highly conserved, tripartite complexes with Veli and Mint1, or Veli and Caskin1 [22, 23]. Liprin-α is another important scaffold protein of the active zone. CASK interacts with liprin-α via the CaMK and first L27 domains of CASK [24–26]. Furthermore, liprin-α interacts directly with the kinesin motor protein KIF1A and is required for the initial localization of CASK to the presynaptic site [27]. Thus, CASK creates a linkage between neurexin and several central scaffold molecules of the active zone. Interestingly, reduced expression of Neurexin leads to an increase in CASK levels in a human disease model [28]. CASK knock-out mice die within the first day after birth. CASK deficient neurons display alterations in spontaneous transmitter release, suggesting that the role of CASK at the presynapse is of central importance [29].

While CASK is predominantly cytosolic in neurons, it can also translocate into the nucleus and act as an effector of neuronal gene expression [30]. This function is mediated by a trimeric complex of CASK with the CASK interacting nucleosome assembly protein (CINAP) and the T-box transcription factor T brain 1 (Tbr1) [30, 31]. The transcription factor Tbr1 is essential for the development of the cerebral cortex [32]. At the post-synapse, CASK fulfils regulatory functions during the transport of α-amino-3-hydroxy-5-methyl-4-isoxazolepropionic acid receptors (AMPARs) and N-methyl-D-aspartic acid receptors (NMDARs). By association with the MAGUK SAP97, CASK modulates the binding affinity of SAP97 to AMPARs and NMDARs in order to regulate the ratio of these receptors at the postsynaptic membrane [33–35].

We asked how these various and seemingly unrelated functions of CASK are regulated. Since alternative splicing is a common mechanism to tweak the function of synaptic proteins dependent on developmental stage, brain region or synaptic compartment, we hypothesized that the diverse functions of CASK could be regulated by alternative splicing events, leading to the expression of different isoforms of the CASK protein. We investigated which transcript

variants of CASK are expressed in the fetal human brain by RT-PCR analysis of fetal human brain RNA, followed by sequence analysis. Six transcript variants, which differed due to the in- or exclusion of four alternatively spliced exons were then analyzed in interaction studies with the known interaction partners neurexin, Veli, liprin-α, Tbr1 and SAP97. We show that the protein sequences encoded by the alternatively spliced exons have the capacity to affect binding to specific interaction partners.

## Material and methods

### Human fetal brain cDNA

Fetal total brain cDNA from a 22 weeks old female (Catalog No.: R1244035-50; LOT#B210035; clinical diagnosis: normal) was obtained from Biochain Institute, Newark, CA, USA.

### Ethics statement

Human fetal cDNA, human cell lines and human DNA clones were obtained from commercial suppliers; therefore no informed consent could be obtained, and no ethics statement is necessary. Work on human subjects in the Institute for Human Genetics has been approved by the Ethics Committee of the Hamburg Chamber of Physicians under approval number PV 3802.

### Expression constructs

The cDNA coding for human CASK transcript variant 3 (TV3; Addgene #23470; contributed by W. Hahn and D. Root, USA) was cloned into a pmRFP-C1 vector using EcoRI/KpnI restriction sites, which lead to the expression of an mRFP-CASK fusion protein carrying an N-terminal mRFP-tag. The primers 5'-TCACCCATGGCTTAAGGAGCGGGATCGTTACGCCT-3' and 5'-GCCTAATAAGACTAGTGTTTTCCTCTTGAATGCTGGCAGT-3' were used to amplify the central part of the CASK coding sequence from fetal human brain cDNA, which is subject to alternative splicing. PCR products in the 1.3 to 1.5 kb range were transferred into the AflII and SpeI sites of the mRFP-CASK TV3 vector by In-Fusion HD cloning Kit (Takara) according to manufacturer's protocol. 29 clones from this experiment were sequenced and analyzed to identify splice variants of CASK expressed in the fetal human brain (Fig 1).

898 bp of 5' upstream region of the human Reelin gene (RELN; GenBank accession number: AC002067; NC_000007.14:g.103990907_103991804) and 769 bp of promotor region of murine NMDA receptor subunit epsilon 2 gene (*Grin2b*; GenBank accession number: AF033356.1; NC_000072.7:g. 136150198_136150965) were amplified from genomic DNA and subcloned into the KpnI and XhoI sites of the luciferase reporter pGL3-basic/enhancer vector (*RELN*: forward primer: 5'-tttggtaccaagataattatacctactttgcaggc-3' and reverse primer: 5'-tttctcgagttcttcctcttactgagaccg-3'; Nr2b: forward primer: 5'-tttggtaccctattgtttaactgtcaatttcctg-3' and reverse primer: 5'-tttctcgagagcccagattccagttggtg-3'). Underlined residues correspond to the KpnI and XhoI restriction sites within the forward and reverse primers, respectively. All constructs were sequenced for integrity.

Epitope-tagged expression vectors for a set of known interaction partners of different domains of CASK were obtained as follows: HA-tagged neurexin-1β (Nrxn1β; a ligand for the PDZ domain of CASK) was obtained through Addgene (catalog #58267) from P. Scheiffele (Basel). HA-tagged liprin-α2, which interacts with the CaMK and L27.1 domains, was obtained from C. Hoogenraad [36, 37]. Myc-tagged Tbr1, which associates with the C-terminal GK domain of CASK [30, 31, 38] was obtained from Y.P. Hsueh (Taiwan).

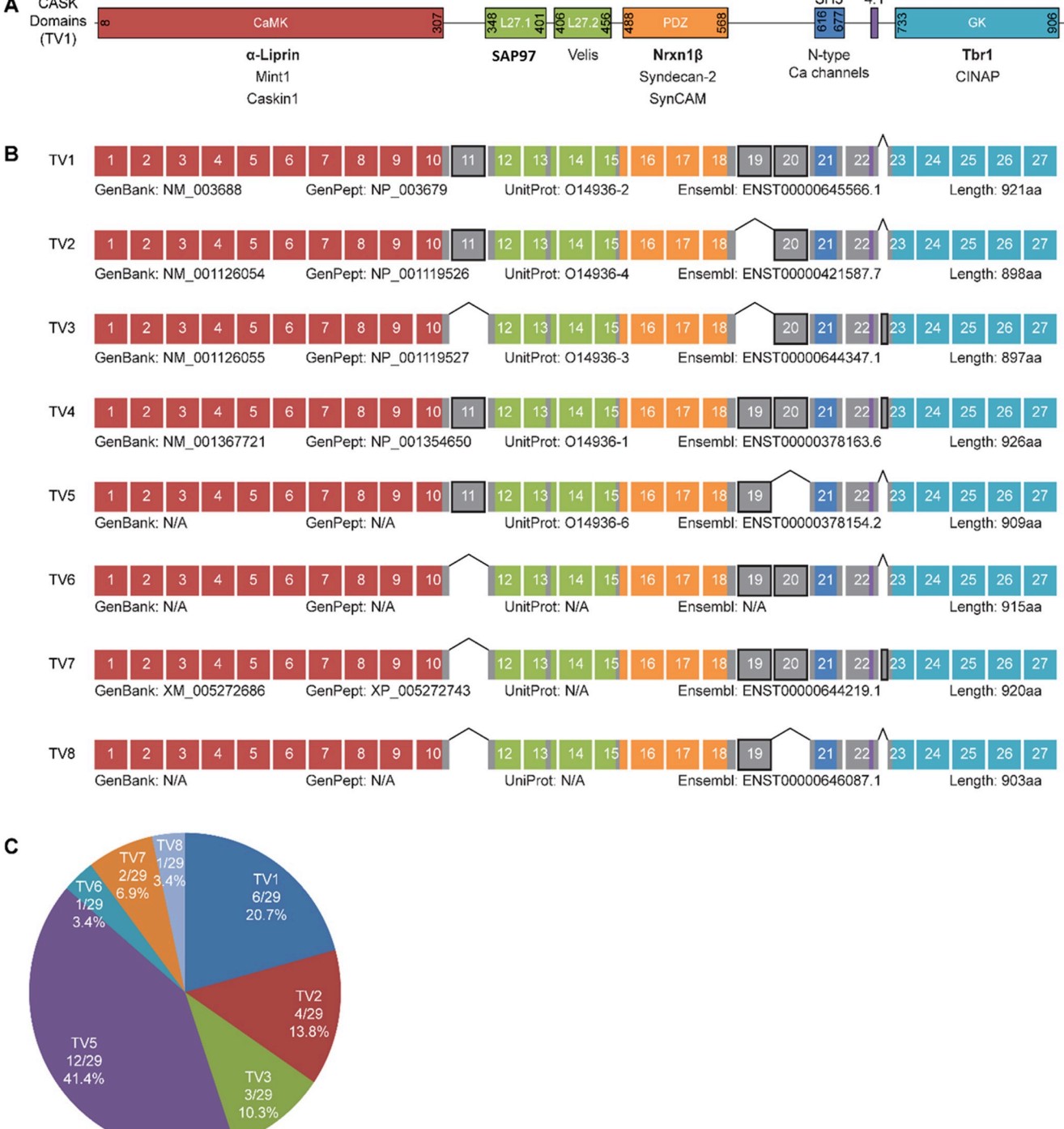

**Fig 1. Variants of CASK expressed in the fetal human brain.** A: The domain structure of the CASK protein with a selection of interaction partners for the respective domains. Interaction partners tested here are shown in bold. B: Result of the analysis of fetal human brain RNA by RT-PCR, followed by subcloning of PCR products and Sanger sequencing shows the exon structure of human variants of CASK. CASK TV4 was not present among the tested clones but added in the figure, as it is listed as known human variant at GenBank. The alternatively spliced exons are indicated in grey and encode for linker regions between defined protein domains. C: Relative abundance of CASK variants among the 29 tested clones. CASK TV5, corresponding to Cask from rat in terms of exon usage, accounted for 41.5% of all transcript variants and was the most abundant variant.

## Antibodies

For application in western blot experiments, dilutions of primary antibodies were prepared in TBS-T with 5% milk powder. The following antibodies were used: α-CASK (rb; 1:1000; Cell Signaling Technologies #9497S); α-HA (ms; 1:1000; Sigma #H9658), α-myc (ms; 1:1000; Sigma #5546); α-Veli 1/2/3 (rb; 1:1000; Synaptic Systems, 184 002); α-SAP97 (ms; 1:500) was provided by Dr. Stefan Kindler (Inst. for Human Genetics, UKE Hamburg). The HRP-coupled secondary antibodies gt-α-rb and gt-α-ms were purchased from ImmunoReagents, Inc. and diluted 1:5000 in TBS-T for use in western blot analysis.

## Cell culture

HEK293-T cells were cultivated on 10 cm dishes in DMEM at 37°C, 5% CO2 and humidified air. Cells with 70% confluence were transferred to new culture vessels. For passaging, cells were first washed with versene and detached by trypsinization. The cells were resuspended in full medium and the respective amount of cell suspension was seated for the required confluence.

## Cotransfection and coimmunoprecipitation

At 40% confluence HEK293-T cells were transiently transfected using TurboFect transfection reagent (Thermo Fisher Scientific). Per plate 4 μg DNA was used for each plasmid. 24 h after transfection the cells were lysed in RIPA buffer with protease inhibitors (1:500 0,125 M PMSF, 5 mg/mL leupeptin, 1 mg/mL pepstatin A) for 15 min at 4°C, followed by centrifugation at 20.000 x g for 15 min. 50 μL of the cell lysate was taken as input sample (IN). All remaining cell lysate was then mixed with RFP-Trap beads (Chromotek) and RFP-tagged CASK was immunoprecipitated. For this, the samples were placed on a rotator for 2 h at 4°C. Afterwards the beads were washed five times with ice-cold RIPA buffer (1000 x g, 4°C, 1 min) and the supernatant was discarded. The beads were resuspended in Laemmli buffer to obtain the precipitate samples (IP). The input and precipitate samples were incubated at 95°C and 200 rpm for 5 min and analyzed by western blotting. The protein bands were detected by chemiluminescence using a BioRad imaging system. For quantification, the recording system was set to "auto" mode which allows for the accumulation of luminescence signal until the first pixel of the image reaches saturation. This mode thus avoids quantification of bands which have been oversaturated. The relative protein amounts of the precipitate samples were then quantified using ImageLab 6.0 (BioRad). In each experiment, a strong increase in CASK IP signal over CASK in the inputs was obtained.

## Dual luciferase assay

HEK293-T cells were seeded on 6-well plates and cultivated for 24 h. Then, the cells were cotransfected to overexpress different CASK variants, Tbr1, a firefly luciferase plasmid under the control of the mouse *Grin2b* or human *RELN* promoter and a renilla luciferase plasmid. 24 h after transfection, the cells were washed twice with cold passive lysis buffer (PLB) and lysed in PLB for 15 min at room temperature (RT). The cell lysates were centrifuged at 20,000 x g and RT for 2 min. Lysates were used for the dual luciferase assay using the Promega Dual-Glo™ Luciferase Assay System (Thermo Fisher Scientific) according to the manufactures protocol.

## Molecular modelling

Structural inspection and alignments, homology modelling and initial loop building of inter-domain linkers have been done with SwissPDBViewer (spdbv.vital-it.ch). Minor positional

refinements were done with FOLDX (foldx.es) and controlled by PROCHECK (ebi.ac.uk), while larger loop fitting and energy minimizations like the HOOK-HOOK model building within the CASK dimer, have been performed using FOLDIT standalone (fold.it). Final figures were rendered using PyMOl (pymol.org). CASK domain structures of CaMK (PDBID: 3TAC), L27.1 (1RSO), L27.2 (1Y74), PDZ (6NID), and GK (1KGD) have been completed to full-length CASK in different conformations using coordinates of PSD-95 (1JXM), ZO-1 (3SHW) and PALS (4WSI).

## Results

### Identification of seven variants of CASK expressed in the fetal human brain

The pre-mRNA encoded by the human *CASK* gene is subject to multiple alternative splicing events; four experimentally verified (TV1-4), as well as eight predicted variants are present in NCBI databases. We analyzed which transcript variants are expressed in the fetal human brain. Three alternatively used exons and an alternative 5'-splice site are present in the central part of the coding sequence, which may affect interactions of the CASK protein with known binding partners. We performed RT-PCR on fetal human brain RNA to produce cDNA of the central coding region of CASK. We subcloned the resulting fragments into a pmRFP-C1 vector which already carried the (non-variable) 5'- and 3' parts of the CASK coding region. 29 clones were analyzed by Sanger sequencing. Using this approach we identified transcript variants TV1-TV3 from the Refseq database; TV4, which is listed as a known human variant in the NCBI database, was not identified in our selection of clones (Fig 1). Four more variants were identified which were labelled TV5-TV8 here. All variants differ due to the in- or exclusion of the three alternatively spliced exons 11, 19, and 20, as well as due to the usage of an alternative 5' splice site of exon 23 (leading to a longer version of exon 23, named 23L from now on). Transcript variants TV5, TV6 and TV8 are not listed in GenBank as known human variants, whereas TV7 is listed as a predicted variant (RefSeq: XM_00527686). By comparing the CASK protein domains with the exon structure of the CASK mRNA we observed that all alternative exons (depicted in grey in Fig 1B) code for parts of the linker regions between the defined protein domains. Exon 11, with a length of 18 bp, encodes for a short insert between the CaMK domain and the N-terminal L27.1 domain. The adjacent exons 19 (69 bp) and 20 (36 bp) encode residues located between the PDZ and SH3 domains. We recognized that no variant of CASK was lacking both of these alternative exons simultaneously. Exon 23L, which encodes an elongated linker sequence between the protein 4.1 binding domain and the GK domain, was only present in two of the variants we found: TV3 and TV7. The frequency of the different CASK variants among the 29 clones differed considerably. TV5, which corresponds to the initial *Cask* cDNA cloned from rat brain [16], was detected most frequently as it accounted for 41.4% of the sequenced clones. TV1, TV2 and TV3 made up for 20.7%, 13.8% and 10.3% respectively, whereas the variants TV6 and TV8 were present only once (3.4%) among the analyzed clones (Fig 1C).

### CASK TV2 is a stronger binder for neurexin-1β than CASK TV1 and TV7

To determine if these variants have distinguishable properties in terms of protein-protein interactions, we chose five known interaction partners of CASK, which associate with different domains of CASK, distributed over the entire length of the protein. Beginning with the PDZ ligand neurexin-1, we performed coexpression/coimmunoprecipitation (CoIP) experiments. HEK293-T cells were transiently transfected with expression plasmids for mRFP-CASK and HA-neurexin-1β. The shorter splice variant of neurexin-1, neurexin-1β (Nrxn1β) used here, shares the identical C-terminal PDZ ligand motif which mediates interaction of CASK with all

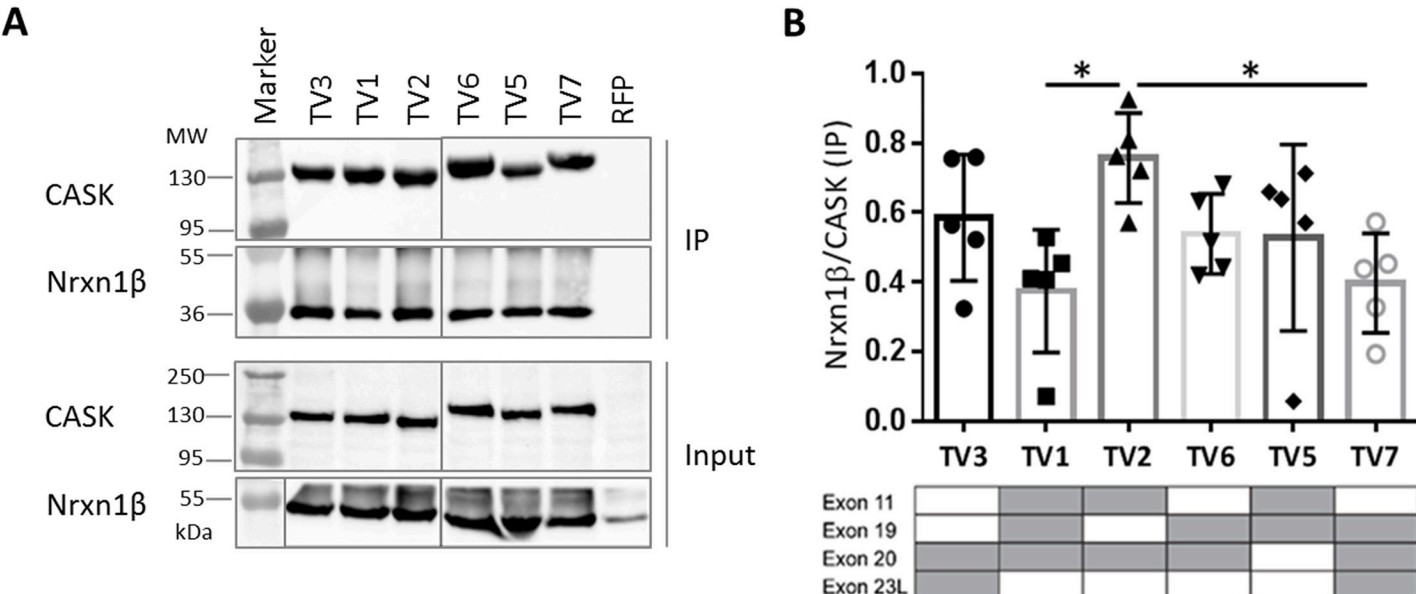

**Fig 2. Effect of alternative splice events on binding of CASK to neurexin-1β (Nrxn-1β).** A: HEK293-T cells expressing mRFP-tagged CASK variants one to seven (TV1-TV7; TV4 not present) or mRFP alone and HA-tagged Nrxn-1β were lysed using RIPA buffer. RFP-tagged CASK was immunoprecipitated by the RFP-Trap matrix and analysed by western blot using α-CASK and α-HA antibodies. All samples were run and analyzed on the same gel/blot, and intervening lanes with non-relevant samples are not shown. B: The ratio of Nrxn-1β/CASK was used as a measure of strength of the interaction. Shown are the means ± SD of five independent experiments. All CASK isoforms precipitated neurexin efficiently. CASK TV2 bound Nrxn1β more efficiently than CASK TV1 and TV7. The presence of alternatively spliced exons is indicated by grey colour in tables below the respective bar graphs. *, P≤0.05; one-way ANOVA followed by Tukey's multiple comparisons test, n = 5.

neurexin isoforms [16]. The mRFP-tagged CASK variants, as well as mRFP only serving as negative control, were immunoprecipitated from cell lysates using the RFP-Trap matrix, and input and precipitate samples were analyzed by western blot using epitope-specific antibodies (Fig 2A). The efficiency of coprecipitation, determined here as the ratio of immunoprecipitated Nrxn1β to CASK, was used as a measure of strength for the interactions. Using this approach, we found that all tested variants of CASK were able to efficiently and specifically coprecipitate the coexpressed Nrxn1β. CASK variant TV2 showed particularly strong binding which was statistically significant when compared with TV1 and TV7 (Fig 2B).

## All CASK variants bind equally well to Veli proteins

CASK is part of a highly conserved tripartite complex together with Mint1 and Veli proteins [23]. For testing the interaction with Veli proteins, we took advantage of the fact that two Veli variants are highly expressed endogenously in 293-T cells. Thus, we transfected cells with vectors coding for the different mRFP-tagged variants of CASK (or mRFP alone) and precipitated the expressed proteins from cell lysates. Strong and specific coprecipitation of Veli proteins (two bands at about 26 and 30 kDa, corresponding to Veli1 at 30 kDa and Veli2/3 at 26 kDa) was detected for all splice variants of CASK, indicating that alternative splicing does not affect binding to Velis.

## Binding to liprin-α2 is weaker with CASK TV5 but stronger with TV7

Liprin-α2 interacts with the N-terminal part of CASK, where binding is mediated by the CaMK and L27.1 domains [26, 36, 39]. The functional relevance of alternative splice events in CASK for binding to liprin-α2 was analyzed by coimmunoprecipitation. A strong interaction with liprin-α2 was observed for all variants tested. When compared to the majority of CASK

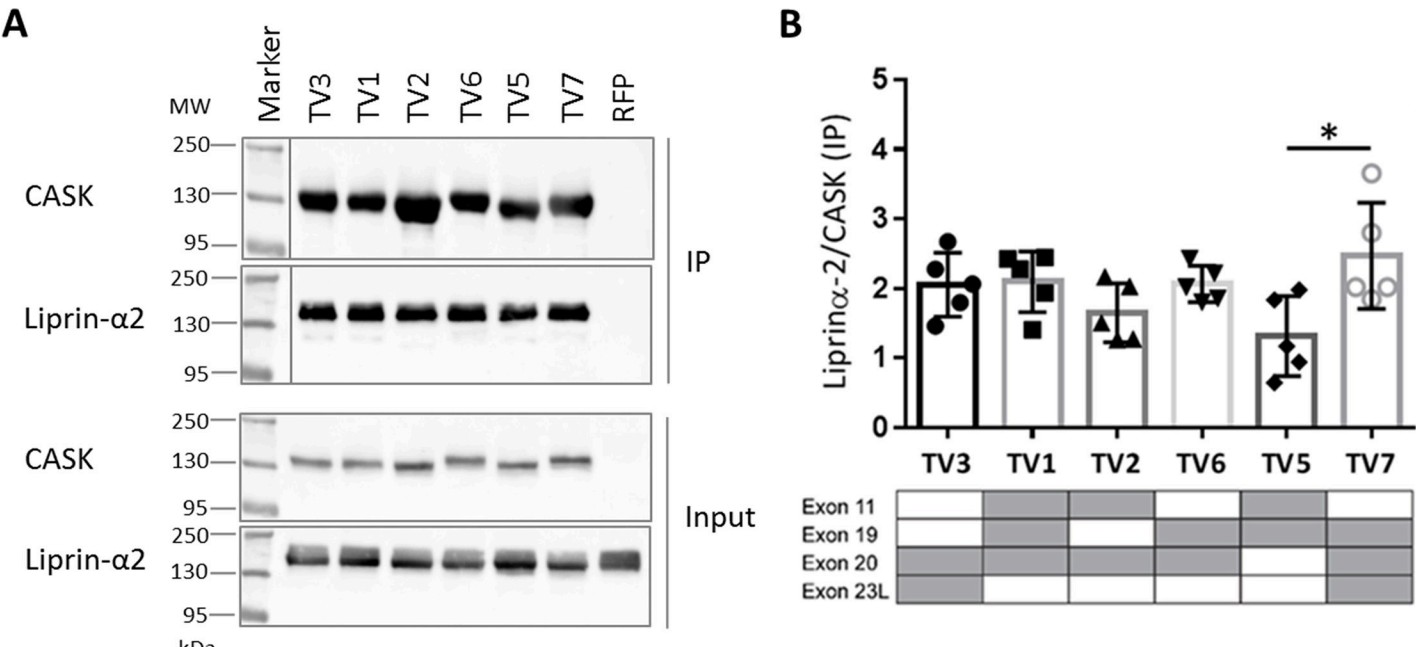

**Fig 3. CASK TV5 binds liprin-α2 weaker than TV7.** A: HEK293-T cells were transfected with plasmids encoding for different RFP-tagged CASK variants (TV1-TV7; TV4 not present), as well as HA-liprin-α2. RFP was used as a negative control. After lysis of the cells in RIPA buffer, RFP-tagged proteins were immunoprecipitated by RFP-Trap. In western blot analysis, CASK and liprin-α2 were detected in input and immunoprecipitate (IP) sample using α-CASK and α HA antibodies. B: Liprin-α2 IP signals were normalized against the appropriate CASK IP signal in five independent repeats. The means ± SD are shown. The presence of alternatively spliced exons is indicated by grey colour in tables below the respective bar graphs. * P≤0.05, one-way ANOVA with Tukey's multiple comparisons test, n = 5.

variants, RFP-tagged CASK TV5 precipitated less liprin-α2, suggesting a slightly weaker interaction. On the other hand, TV7 exhibited a stronger interaction to liprin-α2 (Fig 3B). Although we did not observe any clear correlation between usage of one alternative exon and binding strength of CASK to liprin-α2 in particular, an effect of alternative splice events on liprin-α2-CASK binding was shown.

## Tbr1 shows a weak preference for CASK TV2

We tested the interaction of Tbr1 with CASK variants using coimmunoprecipitation experiments. This assay indicated that TV2 acts as a slightly stronger binder towards Tbr1, though this difference did not become significant after four experiments (Fig 4B). CASK and Tbr1 can form a complex with CINAP to mediate transcriptional activity (31). We attempted to correlate binding of CASK variants to Tbr1 with an effect on Tbr1-mediated transcriptional activity. Following up our finding that CASK TV2 is the favored binding partner for Tbr1 among our tested CASK variants, we hypothesized that this variant could lead to stronger promotor activity than the remaining variants. In dual luciferase assays, using reporters driven by the promoter of *Grin2b* (which encodes the NR2B subunit of NMDA receptors) or *RELN* (which encodes Reelin, a major factor in neuronal differentiation), we observed an increase in promoter activity upon expression of Tbr1, as described by ref. [31]. However, this was not increased further by coexpression of any of the CASK transcript variants (Fig 4C and 4D). This precluded any further validation of the transcript variants in this type of assay.

As it was not clear why we could not increase promoter activity by coexpression of CASK with Tbr1, we analyzed whether Tbr1 was able to induce recruitment of CASK to cellular nuclei. This was indeed the case, as we observed that RFP-CASK (and RFP alone) were present

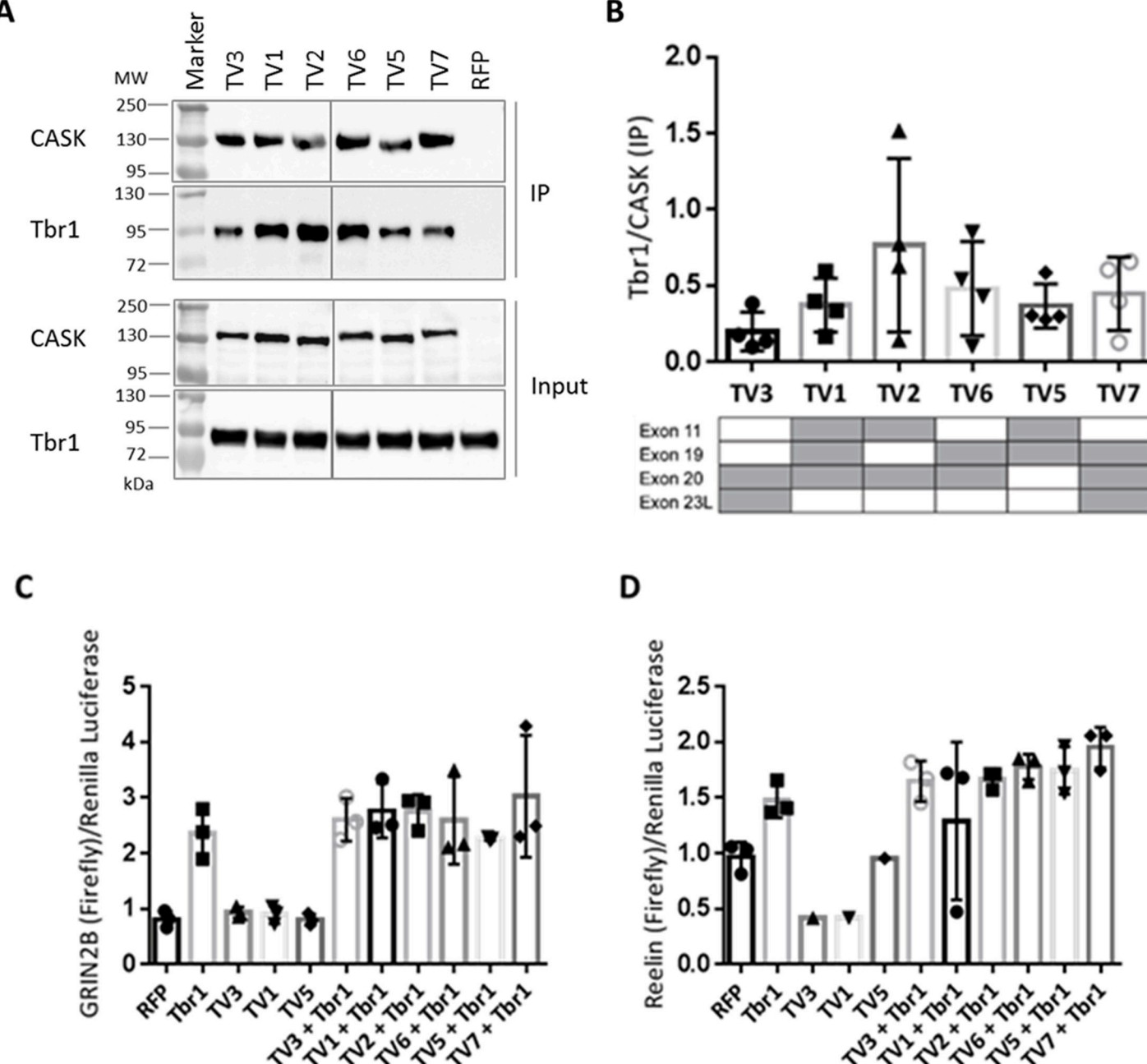

**Fig 4. CASK TV2 is the preferred binding partner of Tbr1.** A: HEK293-T cells expressing mRFP-tagged CASK variants (TV1-TV7; TV4 not present), or mRFP as a negative control, in combination with Myc-tagged Tbr1 were lysed in RIPA buffer 24 h after transfection. Using a RFP-Trap matrix RFP-CASK as well as bound Tbr1 were immunoprecipitated. The input and precipitate (IP) samples were analyzed by western blot, followed by detection with specific α-CASK and α-Myc antibodies. The protein bands were visualized by chemoluminescence using an imaging system (BioRad). The presents of alternatively spliced exons are indicated by grey colour in tables below the respective bar graphs. All samples were run and analyzed on the same gel/blot, and intervening lanes with non-relevant samples are not shown. B: CASK and Tbr1 IP bands were quantified and Tbr1 signals were normalized to CASK signals as a measure of binding strength of each CASK variant. Four independent experiments were performed and the means ±SD are shown. CASK variant two (TV2) bound stronger to Tbr1 compared to the remaining variants, though this difference did not reach significance after four repeats. One-way ANOVA with Tukey´s multiple comparison test. C: HEK293-T cells expressing firefly luciferase driven by the *GRIN2B* promotor, Renilla luciferase under control of a CMV promotor and Tbr1 and CASK variants as indicated were lysed and the activity of both luciferases was determined using a dual luciferase assay kit. For quantification, the ratio of firefly to Renilla signal intensities was calculated for each condition. Upon Tbr1 expression, an increase of promotor activity was induced, which could not be increased further by coexpression of any CASK variant. D: Luciferase reporter assays were performed as in C, using the firefly luciferase expression under control of the *REELIN* promoter.

in the cytosol of transfected 293-T cells. Tbr1 alone was found in nuclei of transfected cells. Upon coexpression, both Tbr1 and RFP-CASK were colocalized in the nuclei of transfected cells (Fig 5), indicating that the interaction with Tbr1 indeed leads to targeting of CASK to the nucleus, as described [30, 31].

## Exon 11 interferes with binding of CASK to SAP97

Similarly, we analyzed the interaction of CASK variants with the postsynaptic scaffold protein SAP97, which is mediated by L27.1 in CASK, and by the single L27 domain of SAP97 [33]. These experiments revealed that the CASK variants TV3, TV6 and TV7 are strong binding partners for SAP97, whereas the remaining variants of CASK (TV1, TV2 and TV5) showed significantly weaker binding (Fig 6). By comparing the exon structure of the CASK variants (Fig 1B) with the results of the binding experiments (Fig 6B), we recognized that the lack of exon 11 had a profoundly positive effect on binding of CASK to SAP97. The inserted amino acid sequence encoded by exon 11 is located adjacent to the L27.1 domain (Fig 1B) which binds exclusively to SAP97. The presence of this insertion reduces the binding capacity of SAP97 to CASK by up to 50% (Fig 6A and 6B; TV1, TV2 and TV5).

To confirm these data, we made use of the endogenous SAP97 expressed in 293-T cells. We used the same samples as shown in Fig 7, where mRFP-CASK variants only had been transfected and immunoprecipitated. These samples were analyzed using a SAP97-specific antibody. Here again we observed the same pattern as in Fig 6, showing that CASK variants lacking exon 11 bound much better to SAP97 than TV1, 2 and 5 which include exon 11 (Fig 8).

In Fig 9 the structure of the L27.1 complex of CASK and SAP97 with the splice insert encoded by exon 11 is shown. The inserted residues (LLAAER, depicted in magenta) extend the first helix of L27.1 in CASK; out of these, only the second alanine has direct contact to SAP97, whereas the others do not contribute to the interaction (Fig 9A). Without these residues the binding capacity of CASK to SAP97 is nearly doubled (Figs 7B and 8B). This argues for a conformational restriction caused by the splice insert. In the absence of exon 11, the rather flexible sequence TSSG replaces this helical extension. This may allow for a better positioning of L27.1 towards L27 of SAP97 (Fig 9A). This would stabilize the CASK-SAP97 complex and result in stronger binding of CASK TV3, TV6 and TV7 to SAP97.

## Discussion

The alternative splicing of CASK, as well as its effects on CASK interactions are aspects of CASK-related research which have not been addressed thoroughly so far. In this study, we identified seven splice variants of CASK expressed in the fetal human brain which differed due to the in- or exclusion of three alternative exons and usage of an alternative 5'-splice site. We outlined that CASK isoforms show functional preferences in terms of protein-protein interactions for the interaction partners neurexin-1β, Veli, liprin-α2, Tbr1 and SAP97. In respect to SAP97 interaction we characterized two groups of CASK variants: the strong binder TV3, TV6 and TV7 and the weak binder TV1, TV2 and TV5. These findings are supported by molecular modeling studies that highlight a direct influence of the splice insert encoded by exon 11 on SAP97 binding affinity. We hypothesize that alternative splicing could be a regulating mechanism to guide the various functions of CASK.

Initially we noted that only TV1 –TV4, but not the other splice variants identified here are annotated in GenBank; also not all variants were represented in the Ensembl list of transcripts, with some transcripts changing their accession numbers during the course of this project. Fig 1 lists all current accession numbers across different databases. CASK TV7 was only listed as a

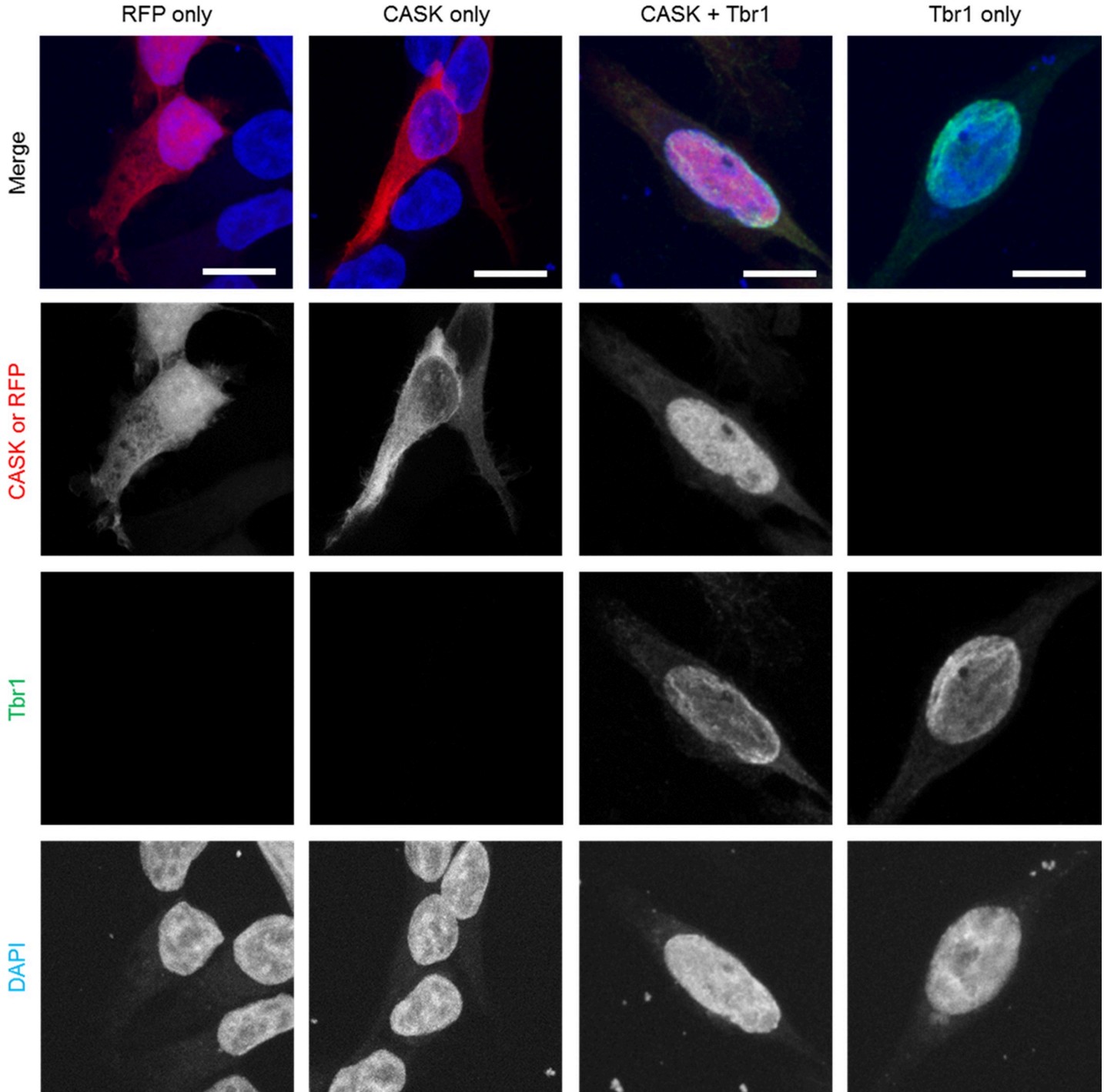

**Fig 5. CASK localization switches from cytoplasmic to nuclear in the presence of Tbr1.** HEK293-T cells were transfected with plasmids encoding a combination of RFP, RFP-CASK-WT and or Myc-Tbr1. Cells were fixed one day after transfection and fixed with paraformaldehyde. Myc and cellular nuclei were stained using α-Myc antibody and DAPI, respectively. Cells were imaged on a SP8 confocal microscope. Overexpressed RFP is present both in the cytoplasm and nucleus of the cells (column 1) while CASK is present primarily in the cytoplasm (column 2). However, in the presence of Tbr1, CASK localization changes almost entirely to the nucleus (column 3). Tbr1 is localized in the nucleus both with (column 3) and without CASK (column 4).

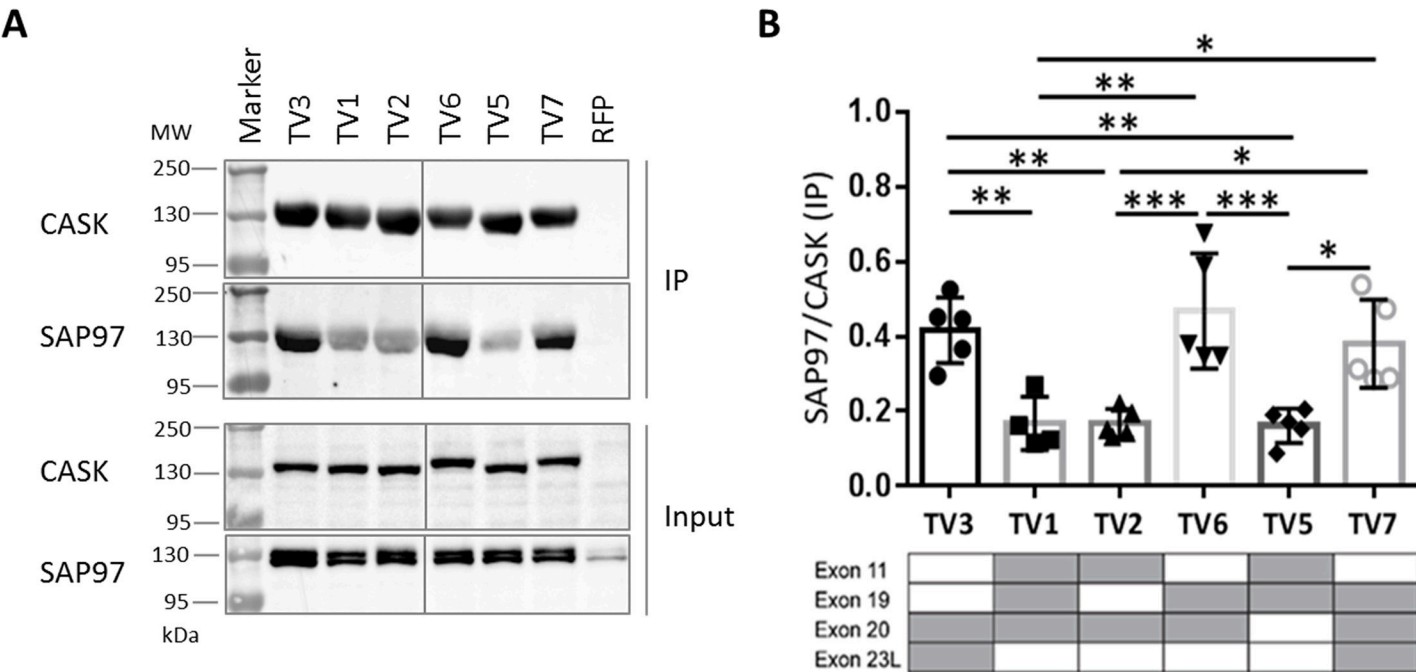

**Fig 6. The absence of splice insert encoded by exon 11 in CASK variants results in stronger SAP97 binding.** A: HEK293-T cells expressing RFP-CASK variants (TV1-TV7; TV4 not present) or RFP together with Myc-tagged SAP97 were lysed in RIPA buffer, and RFP-tagged CASK proteins were immunoprecipitated by RFP-Trap. Input and immunoprecipitate (IP) samples were analysed by Western Blot, using α-CASK and α-Myc antibodies. All samples were run and analyzed on the same gel/blot, and intervening lanes with non-relevant samples are not shown. B: The binding affinity of each CASK variant was determined as ratio of SAP97 over CASK IP. The presence of alternative exons is indicated by grey colour in tables below the respective bar graphs. Note that all CASK variants lacking the splice insert encoded by exon 11 showed a significantly increased binding strength to SAP97, compared to the variants without splice insert 11. Data are shown as means +/-SD. *P≤0.05, ** P≤0.01, *** P≤0.001, One-way ANOVA followed by Tukey's multiple comparisons test with n = 5.

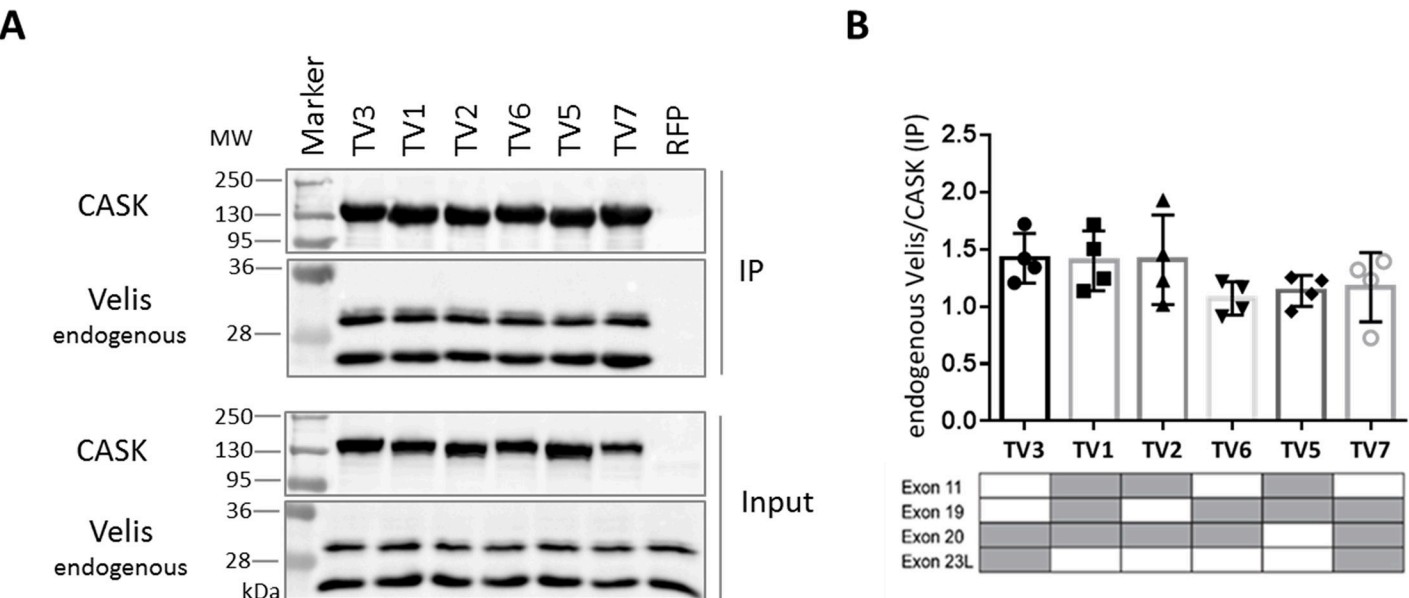

**Fig 7. All CASK variants bind equally well to Veli proteins.** A: HEK293-T cells expressing mRFP-tagged CASK variants or mRFP alone were lysed using RIPA buffer. RFP-tagged proteins were immunoprecipitated by the RFP-Trap matrix; input and precipitate samples were analysed by western blot using α-CASK and α-Veli antibodies. B: The ratio of Veli/CASK signal in precipitates was used as a measure of strength of the interaction. Shown are the means ± SD of four independent experiments. The presence of alternatively spliced exons is indicated by grey colour in tables below the respective bar graphs.

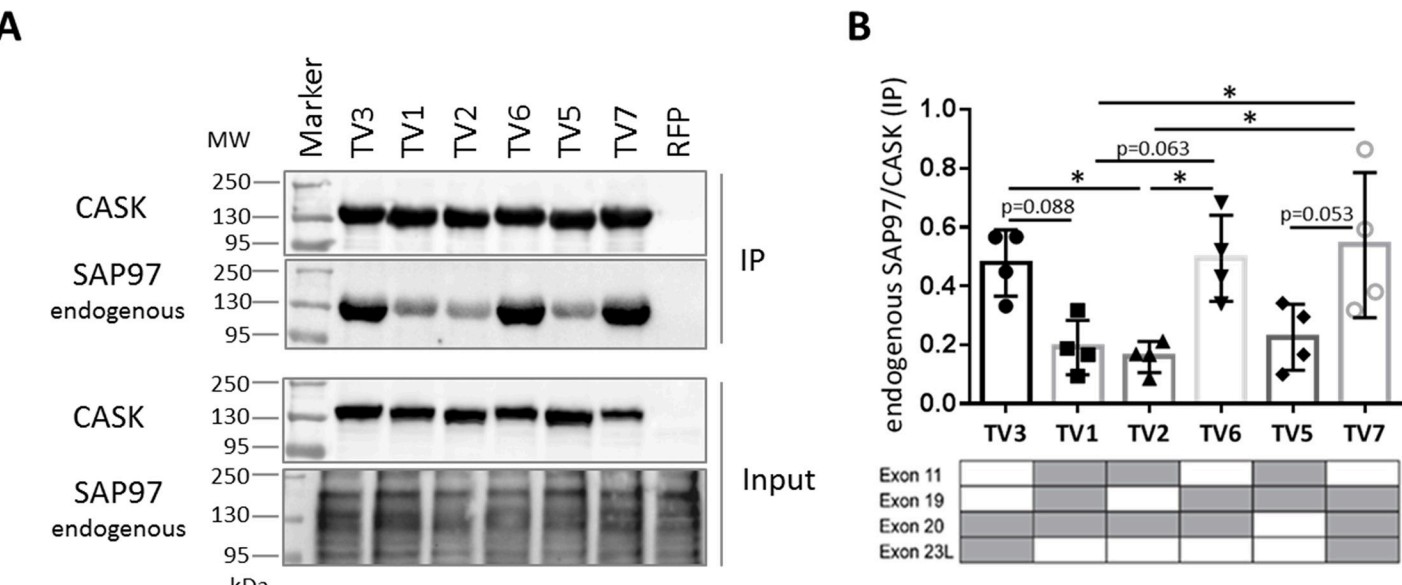

**Fig 8. Interaction of CASK variants with endogenous SAP97.** A. 293-T cells transfected with mRFP-tagged CASK variants, as shown in Fig 3, were subjected to cell lysis and immunoprecipitation of mRFP-tagged proteins. In addition to staining for CASK, input and precipitate samples were analyzed using anti-SAP97. B: Quantification of results, as the ratio of SAP97/CASK signal ration in IP samples. *, $p < 0.05$; one-way ANOVA with Tukey's multiple comparisons test, n = 4.

predicted human variant (XM_005272686.4) while TV6 and TV8 had no match in Genbank. It should also be noted that we did not find CASK TV4 among our 29 clones despite it being a confirmed variant.

Interestingly, TV5 was not present in Genbank; however, TV5 is the equivalent to the rat *Cask* cDNA which was initially cloned by Südhof´s group [16]. As the rat variant has been distributed to many laboratories, it is safe to assume that most functional studies on CASK so far have been performed with the rat sequence corresponding to human TV5.

The relative abundance of variants in our cloning experiment may serve as a first approximation for the abundance of individual variants in fetal human brain. In our hands, TV5 is the most abundant isoform as it accounted for ~40% of our tested clones. Also TV1 was obtained rather frequently (20% of clones). Other variants were in lower abundance, suggesting that these variants are rare.

The statistically more relevant approach to determine the frequency of exon usage and abundance of splice variants is of course provided by large scale RNA sequencing projects as reported e.g. in the Genotype-Tissue Expression (GTEx) and genome aggregation database (GnoMAD) projects. We used the GTEx portal and analyzed the abundance of the alternative exons 11, 19 and 20 in certain brain regions (Table 1). The 5' extension of exon 23 is not listed in GTEx and was excluded from this analysis. The overall relative abundances of the exons 11 and 19 were low compared to exon 20 which was more common in all brain regions. This was surprising and contradicted our analysis, where we observed TV5 (containing exon 11 and 19, lacking exon 20) as the most abundant variant in fetal human brain. However, a closer analysis of sequences spanning exon junctions showed that exon 11 is used in brain and pituitary, but is absent from most other human tissues. Interestingly, expression of exon 11 is increased in the cerebellum compared to other brain regions.

We tried to model the effect of alternatively spliced inserts on CASK protein structure, and on protein interactions of CASK (Fig 9B). The full-length structure of MAGUKs may appear in several conformations, ranging from elongated (pearl-on-a-chain-like) to compact forms

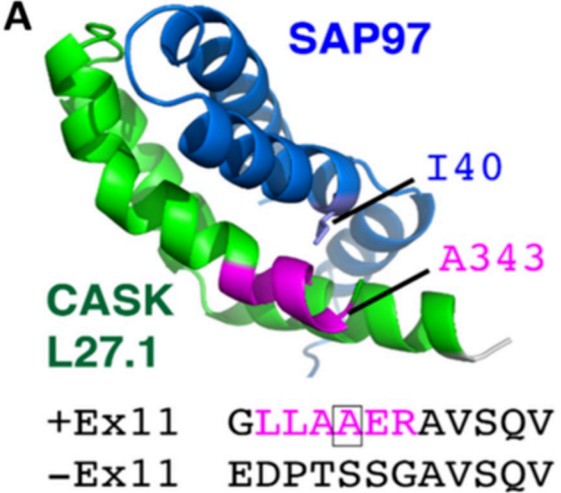

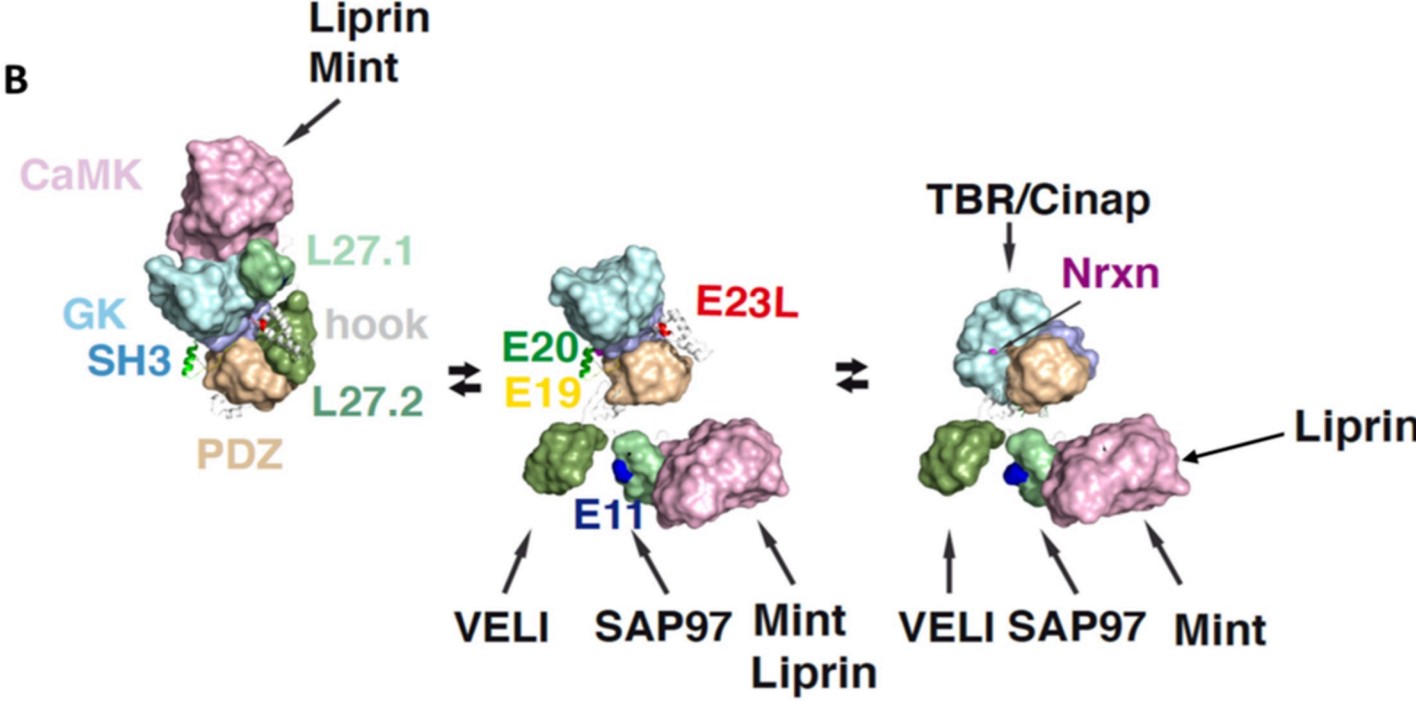

**Fig 9. Effect of alternative splice inserts on structural features of CASK.** A: Role of the splice insert encoded by exon 11 in the CASK-SAP97 complex. The structure of CASK L27.1 in complex with the single L27 domain of SAP97 contains the additional residues of the splice insert encoded by exon 11 (magenta). The splice insert extents the first helix in L27.1 of CASK, allowing for a hydrophobic interaction of A343 in CASK with I40 of SAP97. A lack of this insert brings highly flexible residues like P, G, T and S to this position, likely giving that helix-turn-helix cluster a higher degree of freedom to stabilize this conformation. B: Possible conformational changes of CASK domains, as suggested by structural analysis of other MAGUKs. A compact conformation (left), where CaMK and L27 domains attach to an extended PDZ-SH3-GK module (PSG) and all splice inserts are buried within the complex would only allow binding to CaMK. An extended PSD-95 like conformation (middle) would allow binding to most CASK binding partners. A PALS-like conformation with an U-formed PSG module (right) is required to bind to TBR/Cinap and the cytosolic C-terminus of Nrxn (middle) would allow binding to most CASK binding partners. Note that Mint and liprin compete for the same binding site in the CaMK domain; however, in the presence of neurexin, liprin binds to an alternative site, allowing for the formation of a tetratrameric complex (Nrxn, CASK, Mint and liprin), as described in ref. [26].

**Table 1. Usage of exons 11, 19 and 20 in CASK transcript variants sorted by brain regions according to the Genotype-Tissue Expression (GTEx) database.**

| | Exons | | |
|---|---|---|---|
| **Brain regions** | 11 | 19 | 20 |
| Cerebellum | **0,084** | **0,025** | **0,149** |
| Substantia nigra | **0,000** | **0,008** | **0,179** |
| Putamen (basal ganglia) | **0,000** | **0,009** | **0,140** |
| Caudate (basal ganglia) | **0,013** | **0,013** | **0,204** |
| Hippocampus | **0,002** | **0,013** | **0,175** |
| Amygdala | **0,010** | **0,011** | **0,185** |
| Hypothalamus | **0,039** | **0,022** | **0,218** |
| Nucleus accumbens (basal ganglia) | **0,026** | **0,018** | **0,171** |
| Frontal Cortex (BA9) | **0,041** | **0,019** | **0,192** |
| Cortex | **0,027** | **0,016** | **0,169** |
| Anterior cingulate cortex (BA24) | **0,026** | **0,019** | **0,186** |

Abundance of the alternatively spliced exons 11, 19 and 20 in certain brain regions. Exon 23L is not mentioned as a separate exon fragment and therefore not analyzed here. While exon 20 was strongly expressed in all brain regions, exon 11 and exon 19 were less abundant. Most exons were used in similar abundances in all brain regions with the exception of the cerebellum. There, the expression of variants with exons 11 is increased compared to the remaining brain regions.

[40–42]. The domains rearrange remarkably while switching from the elongated to a compact conformation [42–44]. The necessary flexibility for this process is provided by the unstructured linker regions between the defined protein domains. In CASK, these linker regions are altered by alternative splicing events involving exons 11, 19, 20 and 23L, suggesting that alternative splicing regulates the conformational flexibility of CASK domains, and their orientation towards each other (Fig 9B).

All MAGUKs, except of the MAGI subfamily, share a common domain organization of the C-terminal PDZ-SH3-GK domains known as the PSG tandem module [41, 42]. The PSG module can regulate the specificity as well as the binding affinity of MAGUKs towards their ligands. Dependent on ligand binding the PSG tandem module can be in U-shape or an elongated conformation. Without ligand the protein is elongated (left in Fig 9B), whereas binding of a ligand induces reorganization of the domains to a U-shape [42]. The required flexibility for this conformational change is provided by a hinge or linker region [10, 45]. The splice inserts encoded by exon 19, 20 and 23L are located between the PDZ and SH3 domain (exon 19 and 20) and at the beginning of the so-called hook region in between the SH3 and GK domain (exon 23L; see central model in Fig 9B), thus inside the PSG tandem module. Possibly, the splice inserts affect the binding properties of the PSG tandem resulting in altered binding strengths to interaction partners. No CASK variant was observed that lacked both exon 19 and 20. Since these exons encode for the PDZ-SH3 linker, we speculate that a variant without both splice inserts would not have the sufficient flexibility to form a PSG tandem module, therefore being a non-functional variant (Fig 9B).

Alternative splicing of CASK may be variable during developmental stages or in between different cell populations, as well as dependent on neuronal activity. In murine neurons, skipping of either exon 19 or 20 was inducible by KCl treatment, mimicking neuronal excitation [46]. These data point towards the possibility of the expression of distinct CASK variants dependent on neuronal activity. No variant lacking both exons was detected in that study [46] which matches the observations made here.

We analyzed the effects of the in- and exclusion of splice inserts on the interaction of CASK with known interaction partners. To do so, we chose neurexin-1β, liprin-α2, Tbr1 and SAP97 because they interact with different domains of CASK which are spread over the whole length of the protein. In addition to these recombinantly expressed proteins, we also analysed binding to Veli and SAP97 proteins expressed endogenously in the 293-T cells. We tested these interaction partners to address the question whether distinct variants of CASK could perform specialized functions. In neurons CASK acts as a presynaptic scaffold linking neurexins to components of the active zone, such as liprin-α2 [16, 24–26] and Mint1 and Veli proteins [22, 23]. In a nuclear complex with Tbr1 and CINAP, CASK can regulate transcription, while the interaction of SAP97 and CASK influences the ratio of NMDARs to AMPARs at the postsynaptic membrane [31, 33–35].

We found that binding of CASK to neurexin was consistent among all tested variants except CASK TV2, which coprecipitated slightly more Nrxn1β than TV1 and TV7. In case of the CASK-liprin-α2 interaction we showed that CASK TV5 was rather a weak binder and TV7 was a stronger binder while the remaining variants were having intermediate binding affinities to liprin-α2.

One might ask why variations in the C-terminal part of the protein (such as inclusion of exons 20 or 23L) may lead to altered binding for N-terminal interaction partners such as liprin-α. Work by LaConte et al. [26] provides some important clues here. They showed that binding of Nrxn1 (which we now know involves the whole PSG motif) affects the differential interaction of the CASK N-terminus with Mint1 and liprin-α. Mint1 and liprin-α compete for the same binding site on the CaMK domain in the absence of neurexin. In the presence of neurexin, both Mint1 and liprin-α can bind to CASK without any steric hindrance. Thus, as also suggested in Fig 9, in a folded conformation of CASK the C-terminal (PSG) domains may influence binding at the N-terminal CaMK domain. As a consequence, a variant lacking exon 23L can have an effect on the N-terminal part of the protein, depending on the combination of exons present.

In case of the interaction with Tbr1, only CASK TV2 showed increased binding to Tbr1 which however did not become significant compared to other variants.

For analysis of interaction partners of the two L27 domains of CASK, we took advantage of endogenously expressed interaction partners SAP97 and Veli, in addition to overexpressed SAP97. We did not detect differences between splice variants with respect to Veli binding. In agreement with this observation, Veli proteins bind to the L27.2 domain [33] which is not affected by alternative splicing (see Fig 1). In contrast, we did observe differences between variants with respect to the interaction with SAP97, which binds to L27.1 [33] for both the endogenous, and the overexpressed protein. Here, the variants can be separated into two groups: the weak binding CASK variants TV1, TV2 and TV5 and the strong binders for SAP97, consisting of CASK TV3, TV6 and TV7. The distinguishing feature was the presence or absence of the splice insert encoded by exon 11 (which codes for six residues immediately N-terminal to L27.1). Supported by molecular modeling we demonstrate that alternative splicing of CASK directly influences the conformation of CASK in a complex with SAP97, which resulted in low binding when exon 11 is present. Interestingly, all peripheral tissues appear to express variants deficient in exon 11, consistent with the essential role of the CASK/SAP97 interaction in positioning of the mitotic spindle during generation of epithelial cell polarity [47]. In contrast, endocrine cells appear to use an exon 11 containing variant, as skipping of this exon in mouse islets alters the ability of the Cask protein to affect insulin secretion, supporting the functional relevance of this alternative splicing event [48]. Interestingly, our sample of brain cDNAs indicates that all of the most abundant variants in brain (TV1, TV2 and TV5) are variants

containing exon 11 and therefore are not likely to participate in the regulation of postsynaptic AMPAR/NMDAR ratios (as described by [34, 35]), due to their poor binding to SAP97.

## Supporting information

**S1 File. Original blots.**
(PDF)

## Acknowledgments

We thank Hans-Hinrich Hönck (Institute for Human Genetics, UKE Hamburg) for excellent technical assistance. We also thank Stefan Kindler (Hamburg) for the SAP97 antibody, Peter Scheiffele (Univ. of Basel, Switzerland) for the pCAG-HA-rat Nrxn1β AS4(-) expression construct (obtained through Addgene, catalog #58267) and Yi-Ping Hsueh (Taiwan) as well as Simon Fisher (Nijmegen, Netherlands) for Tbr1 expression vectors. We thank the UKE microscopic imaging facility (umif) for providing confocal microscopes.

## Author Contributions

**Conceptualization:** Debora Tibbe, Yingzhou Edward Pan, Hans-Jürgen Kreienkamp.

**Funding acquisition:** Hans-Jürgen Kreienkamp.

**Investigation:** Debora Tibbe, Yingzhou Edward Pan, Carsten Reißner, Frederike L. Harms.

**Project administration:** Hans-Jürgen Kreienkamp.

**Supervision:** Hans-Jürgen Kreienkamp.

**Validation:** Hans-Jürgen Kreienkamp.

**Visualization:** Carsten Reißner.

**Writing – original draft:** Debora Tibbe.

**Writing – review & editing:** Hans-Jürgen Kreienkamp.

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
