## [Decision Letter · Decision Letter 0]

16 Apr 2021

PONE-D-21-08478

Functional analysis of CASK transcript variants expressed in human brain

PLOS ONE

Dear Dr. Kreienkamp,

Thank you for submitting your manuscript to PLOS ONE. After careful consideration, we feel that it has merit but does not fully meet PLOS ONE’s publication criteria as it currently stands. Therefore, we invite you to submit a revised version of the manuscript that addresses the points raised during the review process.

Both reviewers have raised issues which need to be considered if the authors plan to submit a revised manuscript. 

We look forward to receiving your revised manuscript.

Kind regards,

Salvatore V Pizzo

Academic Editor

PLOS ONE

Journal Requirements:

Additional Editor Comments (if provided):

Reviewers' comments:

Reviewer's Responses to Questions

**Comments to the Author**

1. Is the manuscript technically sound, and do the data support the conclusions?

Reviewer #1: Yes

Reviewer #2: Partly

2. Has the statistical analysis been performed appropriately and rigorously? 

Reviewer #1: Yes

Reviewer #2: Yes

3. Have the authors made all data underlying the findings in their manuscript fully available?

Reviewer #1: Yes

Reviewer #2: Yes

4. Is the manuscript presented in an intelligible fashion and written in standard English?

Reviewer #1: Yes

Reviewer #2: Yes

5. Review Comments to the Author

Reviewer #1: This is an important study on alternative splicing of CASK. I have only one major and one minor comment.

Major

It is regrettable that the paper focuses on the binding of CASK to PSD95 because it is unknown whether CASK is ever postsynaptic. The only evidence for a postsynaptic localization is from a Morgan Sheng paper, who used the same method in a different paper to also localize LAR-type PTPRs to postsynaptic compartments, which is now known to be highly unlikely. The same lab also had a Nature paper claiming that CASK is transcription factor (cited in the manuscript), which has also not been reproduced. A focus on a true interaction as in the Mint-Veli complex would have been more interesting. But this complex (Butz et al., 1998), which is the only physiologically validated CASK interaction besides CASK binding to neurexins, was not discussed or cited.

Minor

1. In English, comma's are not used to denote decimals - we use periods (see Table 1)

2. I think the authors should cite the original literature. Schreiner 2018 is cited, but the original Ullrich 1995 paper is not. The Agoston et al. CASK KO paper - the only known functional analysis of CASK at the synapse - is not discussed or cited, but the Nature transcription factor paper is even though it has been questioned. Similarly, there is no mention that CASK levels go up in human neurons deficient in neurexin-1 (Pak et al., 2015). There should be a more balanced discussion and citations of what has actually been validated for CASK.

Thomas C. Südhof

Reviewer #2: The manuscript describes the analysis of splice variants of the CASK gene in a 22 weeks human embryo. In addition to identify variants not described previously in the Data base the authors clone some of the splice variants and express them in HEK cells together with known partners such as Neurexin.

None of the variants show a difference between them in the ability to bind these proteins or to recruit them to the nucleus. Except for the binding to SAP97. So, at least for the assays described in the manuscript, all variants seem to be equivalent.

Although the manuscript gives only few positive results is useful data and should be published

Following are my main questions, particularly points 4 and 5.

1- Quantification of the chemioluminiscence: I am not familiar with ImageLab 6.0 so please clarify if the system takes several pictures of the luminescence during its development or if the quantification is obtained using the picture provided. Several of the bands look saturated. If the quantification were not obtained from at least two different exposures they are not

2- Pictures in figure 5 have been taken with different zoom, please provide pictures with the same amplification.

3- Please provide MW markers in all WB

4- Figure 6: HEK cell line expresses SAP97 (DLG1), this is notorious in the input that shows a double line. It is not clear if the endogenous SAP97 is IP as well as the transfected one, actually it seems that the endogenous one is the main IP form. Please clarify.

5- This is very important since Cask and SAP97 show a strong binding it is not clear if some of the IP shown is due to CASK splice variants or to the recruitment through endogenous SAP97. It would be necessary a control with SAP97 deficient cells or knock-out HEK cells.

6- Please comment about the difference in the IP efficiency, compare Figure 3 with Figure 4.

7- To me the molecular model shown in the Figure 7 are not clearly connected to the results. The model and the association to the results could be more clear.

6. PLOS authors have the option to publish the peer review history of their article (what does this mean?). If published, this will include your full peer review and any attached files.

Reviewer #1: **Yes: **Thomas C. Südhof

Reviewer #2: No

---

## [Author Response · Author response to Decision Letter 0]

10 May 2021

Response to Reviewers

Reviewer #1: This is an important study on alternative splicing of CASK. I have only one major and one minor comment.

Response: We thank both reviewers for their interest and their positive and constructive comments, which have helped us to improve the manuscript. We have answered the different requests as detailed below.

Major

It is regrettable that the paper focuses on the binding of CASK to PSD95 because it is unknown whether CASK is ever postsynaptic. The only evidence for a postsynaptic localization is from a Morgan Sheng paper, who used the same method in a different paper to also localize LAR-type PTPRs to postsynaptic compartments, which is now known to be highly unlikely. The same lab also had a Nature paper claiming that CASK is transcription factor (cited in the manuscript), which has also not been reproduced. A focus on a true interaction as in the Mint-Veli complex would have been more interesting. But this complex (Butz et al., 1998), which is the only physiologically validated CASK interaction besides CASK binding to neurexins, was not discussed or cited.

Response: we followed these suggestions by adding several new references (Butz et al. 1998, Tabuchi et al., 2002). We think that our manuscript now more adequately reflects the published literature (also see response to minor point below). Importantly, we have also added a final statement that most CASK variants expressed in brain are unlikely to bind to SAP97 (the only PSD-95 family member analysed here) because of the presence of exon 11. 

Experimentally, we have also now included work on the CASK-Mint-Veli complex, taking advantage of endogenous Veli variants expressed in 293T cells. The expressed CASK splice variants were precipitated from cell lysates, and input and precipitate samples were analysed using anti-Veli. As a result, we observed highly efficient coprecipitation for all splice variants, with no differences observed between variants. These new experiments are shown in the new Fig. 3. 

Minor

1. In English, comma's are not used to denote decimals - we use periods (see Table 1)

Response: Table 1 has been changed accordingly.

2. I think the authors should cite the original literature. Schreiner 2018 is cited, but the original Ullrich 1995 paper is not. The Agoston et al. CASK KO paper - the only known functional analysis of CASK at the synapse - is not discussed or cited, but the Nature transcription factor paper is even though it has been questioned. Similarly, there is no mention that CASK levels go up in human neurons deficient in neurexin-1 (Pak et al., 2015). There should be a more balanced discussion and citations of what has actually been validated for CASK.

We followed these suggestions and have included all relevant citations,mostly in the Introduction but also in the Discussion part of the manuscript.

Thomas C. Südhof

Reviewer #2: The manuscript describes the analysis of splice variants of the CASK gene in a 22 weeks human embryo. In addition to identify variants not described previously in the Data base the authors clone some of the splice variants and express them in HEK cells together with known partners such as Neurexin.

None of the variants show a difference between them in the ability to bind these proteins or to recruit them to the nucleus. Except for the binding to SAP97. So, at least for the assays described in the manuscript, all variants seem to be equivalent.

Although the manuscript gives only few positive results is useful data and should be published

Following are my main questions, particularly points 4 and 5.

1- Quantification of the chemioluminiscence: I am not familiar with ImageLab 6.0 so please clarify if the system takes several pictures of the luminescence during its development or if the quantification is obtained using the picture provided. Several of the bands look saturated. If the quantification were not obtained from at least two different exposures they are not

Response: Chemoluminescence pictures were generated in the “auto” mode of the program; this records luminescence exactly until the time point when the first Pixel in the picture reaches saturation, and then stops. Using this mode we are sure that none of the pictures are overexposed (as so frequently happened with autoradiographic films). We have made this more clear in the Materials and Methods section.

2- Pictures in figure 5 have been taken with different zoom, please provide pictures with the same amplification.

Response: This Figure (current Fig. 6) has now been changed to show the same zoom in all subfigures. 

3- Please provide MW markers in all WB

Response: we have now added MW weight markers in all Western Blots.

4- Figure 6: HEK cell line expresses SAP97 (DLG1), this is notorious in the input that shows a double line. It is not clear if the endogenous SAP97 is IP as well as the transfected one, actually it seems that the endogenous one is the main IP form. Please clarify.

5- This is very important since Cask and SAP97 show a strong binding it is not clear if some of the IP shown is due to CASK splice variants or to the recruitment through endogenous SAP97. It would be necessary a control with SAP97 deficient cells or knock-out HEK cells.

Response to questions 4 and 5: We thank the reviewer for this important hint; in fact, in our initial experiment, we used myc-tagged SAP97 and detected it with anti-myc. Therefore, endogenous SAP97 was not detected and cannot have played a role in our analysis (which is shown in Figure 7 in the new version). But in this revised version, we also took advantage of the endogenous expression of SAP97 and analysed interaction of this endogenous form with CASK splice variants. We found the same pattern as before, i.e. interaction with CASK was strongly reduced in variants containing exon 11. These data have now been included as a new Figure 8.

6- Please comment about the difference in the IP efficiency, compare Figure 3 with Figure 4.

Response: In all experiments we performed, we saw a strong increase in CASK IP signal over CASK in the inputs. In cases where input samples were run on a different blot than IP samples, the “auto” setting of the Imager allowed for longer accumulation of luminescence signal, allowing the signal to appear stronger (this applies e.g. to inputs in the Tbr1 experiment, old Fig. 4, now Fig. 5). If input and IP samples were run on the same gel, luminescence was acquired for a shorter time due to IP samples reaching saturation (this applies to Liprin samples, old Fig. 3, now Fig. 4). This does in no way affect our quantitative analysis. We have added a statement that “In each experiment, a strong increase in CASK IP signal over CASK in the inputs was obtained” to the Materials and Methods section.

7- To me the molecular model shown in the Figure 7 are not clearly connected to the results. The model and the association to the results could be more clear.

Response: We have tried to make this connection more clear by clearly stating which parts of the modelling are mentioned in the Discussion section.

---

## [Decision Letter · Decision Letter 1]

20 May 2021

PONE-D-21-08478R1

Functional analysis of CASK transcript variants expressed in human brain

PLOS ONE

Dear Dr. Kreienkamp,

Thank you for submitting your manuscript to PLOS ONE. After careful consideration, we feel that it has merit but does not fully meet PLOS ONE’s publication criteria as it currently stands. Therefore, we invite you to submit a revised version of the manuscript that addresses the points raised during the review process.

A few issues have been raised by Reviewer 2 which we would appreciate your considering after which the manuscript should be acceptable for publication.

We look forward to receiving your revised manuscript.

Kind regards,

Salvatore V Pizzo

Academic Editor

PLOS ONE

Journal Requirements:

Reviewers' comments:

Reviewer's Responses to Questions

**Comments to the Author**

1. If the authors have adequately addressed your comments raised in a previous round of review and you feel that this manuscript is now acceptable for publication, you may indicate that here to bypass the “Comments to the Author” section, enter your conflict of interest statement in the “Confidential to Editor” section, and submit your "Accept" recommendation.

Reviewer #1: All comments have been addressed

Reviewer #2: All comments have been addressed

2. Is the manuscript technically sound, and do the data support the conclusions?

Reviewer #1: Yes

Reviewer #2: Partly

3. Has the statistical analysis been performed appropriately and rigorously? 

Reviewer #1: Yes

Reviewer #2: Yes

4. Have the authors made all data underlying the findings in their manuscript fully available?

Reviewer #1: Yes

Reviewer #2: Yes

5. Is the manuscript presented in an intelligible fashion and written in standard English?

Reviewer #1: Yes

Reviewer #2: Yes

6. Review Comments to the Author

Reviewer #1: No further comments - the system tells me I have to type at least 100 characters, so I am typing away until I reach 100 characters

Reviewer #2: In general all the questions I asked were answered, however last question about the endogenous Sap97 is now more evident in the sense that also endogenous Veli binds to CASK in all its isoforms. Thus, the discussion should include this finding, why or why not consider endogenous Veli and Sap97in the especificity of the binding to Cask splice variants.

7. PLOS authors have the option to publish the peer review history of their article (what does this mean?). If published, this will include your full peer review and any attached files.

Reviewer #1: **Yes: **Thomas C. Südhof

Reviewer #2: **Yes: **Jimena Sierralta

---

## [Author Response · Author response to Decision Letter 1]

21 May 2021

Reviewer #2: In general all the questions I asked were answered, however last question about the endogenous Sap97 is now more evident in the sense that also endogenous Veli binds to CASK in all its isoforms. Thus, the discussion should include this finding, why or why not consider endogenous Veli and Sap97in the especificity of the binding to Cask splice variants.

Response: we have now added a more detailed Discussion on the relevance of the two L27 domains of CASK (L27.1 and L27.2), and have detailed why we think binding to Veli is not affected by alternative splicing, whereas binding to SAP97 is affected.

---

## [Decision Letter · Decision Letter 2]

1 Jun 2021

Functional analysis of CASK transcript variants expressed in human brain

PONE-D-21-08478R2

Dear Dr. Kreienkamp,

We’re pleased to inform you that your manuscript has been judged scientifically suitable for publication and will be formally accepted for publication once it meets all outstanding technical requirements.

Kind regards,

Salvatore V Pizzo

Academic Editor

PLOS ONE

Additional Editor Comments (optional):

Reviewers' comments:

Reviewer's Responses to Questions

**Comments to the Author**

1. If the authors have adequately addressed your comments raised in a previous round of review and you feel that this manuscript is now acceptable for publication, you may indicate that here to bypass the “Comments to the Author” section, enter your conflict of interest statement in the “Confidential to Editor” section, and submit your "Accept" recommendation.

Reviewer #2: All comments have been addressed

2. Is the manuscript technically sound, and do the data support the conclusions?

Reviewer #2: Yes

3. Has the statistical analysis been performed appropriately and rigorously? 

Reviewer #2: Yes

4. Have the authors made all data underlying the findings in their manuscript fully available?

Reviewer #2: Yes

5. Is the manuscript presented in an intelligible fashion and written in standard English?

Reviewer #2: Yes

6. Review Comments to the Author

Reviewer #2: All my concerns have been addressed, the added sentence is short but clarifies the subject related to the endogenous proteins.

7. PLOS authors have the option to publish the peer review history of their article (what does this mean?). If published, this will include your full peer review and any attached files.

Reviewer #2: **Yes: **Jimena Sierralta J.

---

## [Editor Report · Acceptance letter]

7 Jun 2021

PONE-D-21-08478R2 

Functional analysis of CASK transcript variants expressed in human brain 

Dear Dr. Kreienkamp:

I'm pleased to inform you that your manuscript has been deemed suitable for publication in PLOS ONE. Congratulations! Your manuscript is now with our production department. 

Kind regards, 

on behalf of

Dr. Salvatore V Pizzo 

Academic Editor

PLOS ONE